# CAN KERNEL TRANSFER OPERATORS HELP FLOW BASED GENERATIVE MODELS?

## ABSTRACT

Flow-based generative models refer to deep generative models with tractable likelihoods, and offer several attractive properties including efficient density estimation and sampling. Despite many advantages, current formulations (e.g., normalizing flow) often have an expensive memory/runtime footprint, which hinders their use in a number of applications. In this paper, we consider the setting where we have access to an autoencoder, which is suitably effective for the dataset of interest. Under some mild conditions, we show that we can calculate a mapping to a RKHS which subsequently enables deploying mature ideas from the kernel methods literature for flow-based generative models. Specifically, we can *explicitly* map the RKHS distribution (i.e., approximate the flow) to match or align with a template/well-characterized distribution, via kernel transfer operators. This leads to a direct and resource efficient approximation avoiding iterative optimization. We empirically show that this simple idea yields competitive results on popular datasets such as CelebA, as well as promising results on a public 3D brain imaging dataset where the sample sizes are much smaller.

## 1 INTRODUCTION

A flow-based generative model refers to a deep generative model composed using a set of invertible transformations. While GANs and VAEs remain the two dominant generative models in the community, flow based formulations have continually evolved and now offer competitive performance in applications including audio/speech synthesis Kim et al. (2019; 2020), text to speech Miao et al. (2020), photo-realistic image generation Kingma & Dhariwal (2018), and learning cross-domain mappings Mahajan et al. (2020). An important property of such models is the explicit use of a tractable likelihood function, which enables leveraging maximum likelihood principles during training as well as efficient/exact density estimation and sampling. The formulation is invertible by design but this involves higher memory requirements. For example, permitting the bijective mapping to be expressive enough involves increases in the memory footprint Lee et al. (2020); Kim et al. (2019), an issue that is a focus of several recent results Jacobsen et al. (2018); Chen et al. (2016). Moreover, in these models we need to calculate the inverse and backpropagate through all invertible transformations during training. Calculating the inverse incurs a multiplicative increase in cost, usually as a function of the feature dimension, relative to the calculation of the likelihood, an issue addressed to some extent in Dinh et al. (2017); Kingma & Dhariwal (2018).

At a high level, a flow-based generative model bijectively pushes the data density from a source to a target, i.e., from a known simple distribution to an unknown (may be intractable) data distribution. During training, we seek to learn this bijective mapping by maximizing the likelihood of the mapped

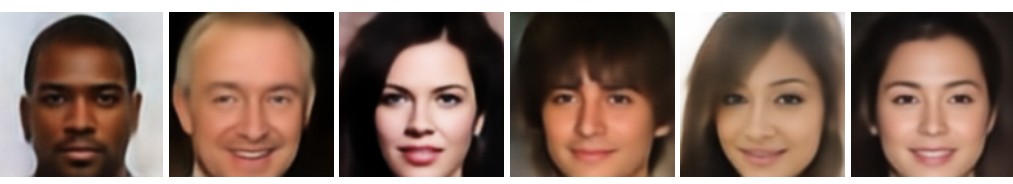

Figure 1: Representative generated images (of resolution $128 \times 128$) using our proposed algorithm.

training samples. In the generation step, we need the inverse of this mapping (given such an inverse exists) to map from a sample drawn from the known distribution back to the input (data) space. When the Jacobian of the transformation mapping can be efficiently computed or estimated (e.g., having a lower triangular form), directly optimizing the likelihood of the training samples is possible. However, in training flow-based generative either we must restrict the expressiveness at each layer or fall back on more numerically heavy solutions, see (Chen et al., 2018). Next, we discuss how several existing results may provide a simplification strategy.

## 1.1 RELATED WORKS AND RATIONALE

Our starting point is the existing literature on Koopman and Perron-Frobenius operators Song et al. (2009); Fukumizu et al. (2013); Klus et al. (2020), that offers an arguably easier, optionally *linear*, procedure that can be used to analyze non-linear dynamics of measurements that evolve temporally. For instance, as described in Arbabi (2018); Lusch et al. (2018), if we view the data/measurements as evaluations of *functions* of the state (of a dynamical system) – where the functions are also called observables – then the entire set of such functions forms a linear vector space. Transfer operators on this space describe a linear evolution of the dynamics, i.e., finite-dimensional nonlinear dynamics are replaced by infinite-dimensional linear dynamics Brunton et al. (2017), perfectly evolving one set of measurements to another over time if the space can be well characterized. Of course, this is not practically beneficial because constructing such infinite-dimensional spaces could be intractable. Nonetheless, results in optimal control demonstrate that the idea can still be effective in specific cases, using approximations with either spectral analysis of large but finite number of functions Williams et al. (2015) or via a search for potential eigenfunctions of the operators using neural networks Li et al. (2017); Lusch et al. (2018). Within the last year, several results describe its potential benefits of such operators in machine learning problems as well Li et al. (2020); Azencot et al. (2020).

If we consider the transformations that flow-based generative models learn as a non-linear dynamics, also used in (Chen et al., 2018), a data-driven approximation strategy one can consider is to map the given data (or distribution) into infinite dimension space of functions through the *kernel trick*, which may allow the use of well known results based on kernel methods, including old and new results on powerful neural kernels Neal (1994); Jacot et al. (2018); Arora et al. (2019). Utilizing these results, a mean embedding on the corresponding Reproducing Kernel Hilbert Space (RKHS) would correspond to the distribution in the input space (the distribution from which input samples are drawn). Therefore, the problem of identifying a nonlinear mapping (or dynamics) in the input space (going from an intractable distribution to a known distribution or vice-versa) reduces to estimating a linear mapping operator between two empirical mean kernel embeddings where recent results on kernel transfer operators Klus et al. (2020) could be relevant or applicable. However, due to the high variability of the data, estimation of the distribution directly in the input space, as we will see shortly, can be difficult. But, if the input space is low-dimensional or otherwise structured, this problem could be mitigated. Fortunately, for many image datasets, one can identify a low-dimensional latent space such that, in theory, the above pipeline could be instantiated, enabling us to learn a transfer operator.

Conceptually, it is not difficult to see how the foregoing idea could potentially help (or simplify) flow-based generative models. In principle, using a transfer operator, one could push-forward the input data distribution to a target distribution of our choice, if both have already been mapped to a sufficiently high dimensional space. If – additionally – the operator could also be inverted, this strategy may, at least, be viable. Of course, several key components are missing. We need to **(a)** assess if setting up a suitable infinite dimensional space is possible, **(b)** identify if we can estimate the transfer operator and then finally, **(c)** check if the procedure works at all. In the following sections of the paper, we will verify these key components and show that using only a linear operator yield surprisingly competitive results on several image generation tasks.

## 2 PRELIMINARIES

**Auto-encoders.** Images often lie close to an (unknown) lower dimensional manifold $\mathcal{M} \subset \mathbf{R}^m$ such that $\dim(\mathcal{M}) \ll m$, and operating with densities in a lower dimensional setting is often much easier.

VAEs leverage this explicitly via auto-encoders. If the parameterized empirical density is $p_\tau(x)$, in VAEs, we write it as $\int p_\tau(x|z)p(z)dz$ where $z$ is the low dimensional representation with a suitable prior. We then use a decoder distribution $q_{\tau'}(z|x)$ and an encoder distribution $p_\tau(x|z)$ and train the parameters $\tau$ and $\tau'$. In practice, $p_\tau(\cdot)$ and $q_{\tau'}(\cdot)$ are assumed to be Gaussian, but it is known that jointly fitting the manifold as well as regressing to a particular prior distribution can be challenging in VAEs Kingma et al. (2016); Dai & Wipf (2019). Now, consider the following approach, where we do not impose a distributional assumption on $z$. If a well-regularized auto-encoder is able to capture information about the data generating density Alain & Bengio (2014), we can think of $z$ as the input measurements (likely, meaningful representations from the input data) which is subsequently mapped to an infinite dimensional space (of observables of these measurements). Now, if we could push-forward the embedded RKHS distribution to the RKHS mapping of a simpler distribution, similar to flow-based generation, one could easily sample from the simple (e.g., standard normal) distribution and transform it via the learned mapping to samples on the latent space. In summary, instead of explicitly searching for the eigenfunctions, we propose to *Step 1:* embed the density from an auto-encoder into a RKHS, *Step 2:* learn a kernel transfer operator in RKHS in one step. In the remainder of this paper, we will provide the details to operationalize this idea and show that this simple approach, in fact, performs surprisingly well with highly favorable computational properties.

We now introduce the definition of reproducing kernel Hilbert space (RKHS) and the kernel embedding of probability distributions which are the building blocks of this paper.

**Definition 1** (RKHS Aronszajn (1950)). *Given a set $\mathcal{X}$ and $\mathcal{H}$ to be a set of functions $\phi : \mathcal{X} \to \mathbf{R}$, $\mathcal{H}$ is called a reproducing kernel Hilbert space (RKHS) with corresponding product $\langle ., . \rangle_\mathcal{H}$ if there exists a function $k : \mathcal{X} \times \mathcal{X} \to \mathbf{R}$ (called a reproducing kernel) such that (i) $\forall X \in \mathcal{X}, \phi \in \mathcal{H}, \phi(X) = \langle \phi, k(X, .) \rangle_\mathcal{H}$ (ii) $\mathcal{H} = \overline{span(\{k(X, .), X \in \mathcal{X}\})}$, where $\bar{.}$ is the completion.*

The *kernel mean embedding* can be used to embed a probability measure in a RKHS.

**Definition 2** (Kernel Mean Embedding Smola et al. (2007)). *Given a probability measure $p$ on $\mathcal{X}$ with an associated RKHS $\mathcal{H}$ equipped with a reproducing kernel $k$ such that $\sup_{X \in \mathcal{X}} k(X, X) < \infty$, the kernel mean embedding of $p$ in RKHS $\mathcal{H}$, denoted by $\mu_\mathcal{H} \in \mathcal{H}$, is defined as $\mu_\mathcal{H} = \int k(X, .)p(X)$, and the mean embedding operator $\mathcal{E} : L^1(\mathcal{X}) \to \mathcal{H}$ is defined as $\mu_\mathcal{H} = \mathcal{E}p$.*

For a characteristic kernel, the mapping from an input space distribution to its corresponding kernel mean embedding is one-to-one. Thus, two distributions in the input space are identical if and only if their kernel mean embedding matches exactly. This property enables *Maximum Mean Discrepancy (MMD)* for distribution matching. For a finite number of samples $\{X_i\}_{i=1}^N$, drawn from the probability measure $\mu$, an empirical estimation of $\mu_\mathcal{H}$ is $\hat{\mu}_\mathcal{H} = \frac{1}{N} \sum_{i=1}^N k(X_i, .)$.

**Definition 3** (Flow-based generative model Dinh et al. (2014)). *A flow-based generative model explicitly learns the data distribution by trying to bijectively map it to a tractable density via invertible transformations. Formally, given a random variable $\mathbf{z}$ following a tractable density, i.e., $\mathbf{z} \sim p_\theta(\mathbf{z})$, a flow-based model learns an invertible mapping $f_\phi$ such that the data sample $\mathbf{x} = f_\phi(\mathbf{z})$ and the corresponding data distribution, $p_{\widetilde{\theta}}(\mathbf{x}) = f_* p_\theta(\mathbf{z})$, where, $f_* = |\frac{dz}{dx}|$ is the push-forward operator.*

## 3 A SIMPLIFICATION STRATEGY FOR FLOW-BASED GENERATIVE MODELS

Flow-based generative model directly learn the mapping from a known distribution to the target data distribution in the input space through likelihood maximization. During image generation, the fact that data may often lie on or near a low-dimensional manifold is usually not explicitly leveraged. If one could estimate this manifold perfectly, we could uniquely identify a distribution in the input space. But often, the manifold can only be empirically estimated. Our simplifying assumption here is that with some structural or smoothness constraints on the distribution on the latent space, we should still be able to approximately identify the true distribution. VAEs use this intuition to make the latent distribution consistent with a standard normal distribution, and several contemporary works also propose estimating flow on latent spaces Kingma et al. (2016); Mahajan et al. (2020). We will follow this rationale and use an autoencoder – here, benefiting from the reduced variance in the latent space, our approach will effectively learn the mapping from a known simple distribution to the empirical latent distribution.

**Structure of the latent space.** We will assume that a suitable auto-encoder is either provided or can be successfully trained from scratch for the data at hand. Given measurements from an input data space, $\mathcal{M}$, we seek to learn a mapping $E : \mathcal{M} \to \mathcal{L} \subset \mathbf{R}^d$, with $d < m$, where $\mathcal{M} \subset \mathbf{R}^m$ and $\mathcal{L}$ is the latent space. Additionally, we also learn the inverse of the mapping $E$ which is denoted by $D : \mathcal{L} \to \mathcal{M}$, i.e., we require $D \circ E = \mathrm{id}$ (identity). Further, we require $\mathcal{L}$ to be a bounded subset of $\mathbf{R}^d$. For simplicity, we model it as a unit hypersphere $\mathbf{S}^{d-1}$. This essentially ensures that $\mathcal{L}$ is **(a)** bounded **(b)** geodesically complete **(c)** geodesically convex space.

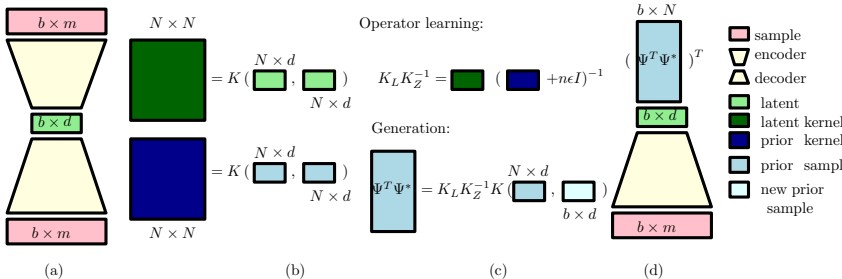

Figure 2: A description of generative procedure in our method. *(a) schematic of an auto-encoder (b) kernel representations of latent space and noise from prior (c) operator learning and inferencing (d) solve pre-image and decode the generated latent representation to obtain final generation result in data space*

*Remark 1.* A bounded latent space with specified structure provides several benefits over one without any such constraints. First, it guarantees that the Gram matrix constructed with a (positive-definite) kernel function is bounded, which is required for a valid empirical estimation of the mean embedding in Def. 2 Song et al. (2009). Moreover, the projection onto the hypersphere also helps ensure the positive-definiteness of kernels constructed using the latent representations, which is discussed in 3.2. At this point, the requirement of geodesic completeness and convexity may not be apparent but these properties will be useful for sample generation.

### 3.1 Using Transfer Operators to estimate Flow

If training the auto-encoder provides us a suitably well-structured latent space, as desired, we can now study how the density transfer operators can help transfer a tractable density to the density on the latent space, i.e., *mimicking* the mechanics within a flow-based generative model.

**Transfer operators.** Let $\{X_t\}_{t \geq 0}$ be a stationary and ergodic Markov process defined on $\mathcal{X}$. Then, we can define the transition density function, $p_\tau$ (with time lag $\tau$), by

$$P[X_{t+\tau} \in \mathcal{A} | X_t = x] = \int_{\mathcal{A}} p_\tau(y|x)dy, \text{ where } \mathcal{A} \text{ is a measurable set on } \mathcal{X}. \tag{1}$$

Let $L^1(\mathcal{X})$ be the space of probability densities. We wish to emulate the dynamical system of interest by instead utilizing a well-known transfer operator, namely the (Ruelle) Perron-Frobenius operator Mayer (1980), where the "transfer" terminology originates from statistical mechanics. Notice that the notion of $t$ in our work is not literal - it merely serves as a convenient way to describe the transfer operators in the context of evolution of states, and is also analogous to how flow is considered explicitly as continuous dynamical systems in Neural ODE (Chen et al. (2018))

**Definition 4** (Perron-Frobenius operator Mayer (1980)). *The Perron-Frobenius (PF) operator $\mathcal{P} : L^1(\mathcal{X}) \to L^1(\mathcal{X})$ push-forwards or transfers a probability density $p_t \in L^1(\mathcal{X})$ given the lag $\tau$ as $(\mathcal{P}p_t)(y) = \int p_\tau(y|x)p_t(x)dx.$*

*Remark 2.* If such an operator can be efficiently estimated, we can use it to transfer the tractable probability density from a known distribution $p_t$ to the target distribution $p_{t+\tau}$ whose density is generally unknown. Since we do not make any distributional assumptions on the target distribution, a linear solution to such dynamics in the input space, in general, may not exist. Nevertheless, in spaces spanned by a sufficiently large set of non-linear functions, or specifically, an RKHS, one can

potentially identify a linear operator that is equivalent to a highly non-linear operator in the input space. In Klus et al. (2020), the authors showed two important results that define such operators in RKHS in terms of covariance and cross-covariance operators, namely kernel Perron-Frobenius operator (kPF) $\mathcal{P}_k$ and embedded Perron-Frebenius operator (ePF) $\mathcal{P}_\mathcal{E}$.

Let $X \sim p_t$ and $Y \sim p_{t+\tau}$ be observations at time $t$ and $t + \tau$. Let $\mathcal{H}$ and $\mathcal{G}$ be the RKHSs associated with a certain kernel $k$, $\phi_H$ and $\phi_G$ be their respective feature mapping. The cross-covariance operator $C_{YX} : \mathcal{H} \to \mathcal{G}$ is defined as $C_{YX} = E_{YX}[\phi_G(Y) \otimes \phi_H(X)]$, and the covariance operator $C_{XX} : \mathcal{H} \to \mathcal{H}$ is defined analogously. If both densities of interest live in the same RKHS, that is, $p_t \in \mathcal{H}$ and $p_{t+\tau} \in \mathcal{H}$, $\mathcal{P}_k$ can be used to push forward the density such that $p_{t+\tau} = \mathcal{P}_k p_t$. However, in our setting, $\mathcal{P}_k$ is not directly applicable since the densities we are interested are elements of $L^1(\mathcal{X})$. Therefore, the densities need to be first embedded into the RKHS and can then be transferred using the $\mathcal{P}_\mathcal{E}$ in the embedded form, namely $\mathcal{E}p_{t+\tau} = (\mathcal{P}_\mathcal{E} \circ \mathcal{E})p_t$.

**Definition 5** (Embedded Perron-Frobenius operator Klus et al. (2020)). *Given $p_t \in L^1(\mathcal{X})$ and $p_{t+\tau} \in L^1(\mathcal{X})$. Let $\mu_t = \mathcal{E}p_t$ and $\mu_{t+\tau} = \mathcal{E}p_{t+\tau}$ be their respective kernel mean embedding. The kernel Perron-Frobenius (kPF) operator, denoted by $\mathcal{P}_\mathcal{E} : \mathcal{H} \to \mathcal{G}$, is defined by $\mu_{t+\tau} = \mathcal{P}_\mathcal{E} \mu_t = C_{YX}(C_{XX} + \epsilon I)^{-1} \mu_t$ (2), under the condition that (i) $C_{XX}$ is injective (ii) $\mu_t \in Range(C_{XX})$ (iii) $E[Y|X = \cdot] \in \mathcal{H}$.*

Notice that the above defined $\mathcal{P}_\mathcal{E}$ essentially has the same form as the kernel conditional embedding in Song et al. (2013). While the first two conditions in the above definition can be satisfied (see Theorem 2 of Fukumizu et al. (2013)), the last condition requires $\exists f \in \mathcal{H}$ s.t. $\forall x$, $f(x) = E[Y|X = x]$, which highlights the importance of the kernel choice. With this operator, we can transfer $p_t$ to $p_{t+\tau}$ in their embedded forms. The following proposition demonstrates the implication of using the embedded Perron-Frobenius operator.

**Proposition 1** (Klus et al. (2020)). *With the above notations, $\mathcal{E} \circ \mathcal{P} = \mathcal{P}_\mathcal{E} \circ \mathcal{E}$.*

*Remark 3.* The commutativity in the proposition shows that the *transferred kernel mean embedding* of $p_t$ by the linear operator we constructed in RKHS is equivalent to the kernel mean embedding of *transferred $p_t$* by a highly nonlinear operator in the input space.

Since $p_{t+\tau}$ is generally intractable, the operator can only be empirically estimated. Given samples $X = \{x_i\}_{i=1}^N \sim p_t^N$ and $Y = \{y_i\}_{i=1}^N \sim p_{t+\tau}^N$, let $\Phi_H$ and $\Phi_G$ denote the feature maps of $X$ and $Y$, respectively. The sample estimate of the embedded Perron-Frobenius operator is given by $\widehat{\mathcal{P}}_\mathcal{E} = \widehat{C}_{YX}(\widehat{C}_{XX} + \epsilon I)^{-1} = \Phi_G(G_{XX} + N\epsilon I)^{-1} \Phi_H^T$ (3) , where $G_{XX} = \Phi_H^\top \Phi_H$ is the Gram matrix of $\Phi_H$. To generate a sample of the transferred density which is *approximately* $p_{t+\tau}$ using a sample $x^*$ of $p_t$, we can construct $\phi_\mathcal{G}(y^*) = \hat{\mathcal{P}}_\mathcal{E} \phi_\mathcal{H}(x_i^*) = \Phi_G(G_{XX} + \epsilon I)^{-1} k(X, x^*)$. The distribution of the resulting samples has the following property,

**Theorem 1** (Proof in A.1). *Let $\mu_{t+\tau}$ be the kernel mean embedding of the true distribution $p_{t+\tau}$. The resulting empirical mean embedding $\hat{\mu}_{t+\tau}^* = \frac{1}{n} \sum_{i=1}^n \phi_G(y_i^*)$ satisfies $E[\hat{\mu}_{t+\tau}^*] = \mu_{t+\tau}$*

Theorem 1 simply implies that $\hat{P}_\mathcal{E} \hat{\mu}_{t+\tau}^*$ is an unbiased estimator of the kernel mean embedding of the true distribution on the latent space. If the kernel is characteristic and the exact preimage exists, then $\phi_\mathcal{G}^{-1}(\hat{P}_\mathcal{E} \phi_\mathcal{H}(x^*)) \sim p_{t+\tau}$ asymptotically, which concludes our main result.

## 3.2    KERNEL TRANSFER OPERATOR FOR SAMPLE GENERATION

We now present an algorithm for sample generation assuming that a pre-trained auto-encoder is available. The detailed algorithm is described in Figure 3. The idea is simple yet powerful: we first generate $n$ samples from a simple distribution restricted to a hypersphere (which we model as a uniform distribution on $\mathbf{S}^{n-1}$). Then, we will construct the operator on RKHS described in the previous section using the sampled points and the latent representations of training samples. At inference time, we will use the operator to transfer new points sampled from the simple distribution to the target feature map. Since it is not practically possible to compute the preimage directly from the infinite-dimensional feature map, we left multiply with $\Phi_G$ and use the geodesic interpolation (gI) module to construct an approximate preimage. Finally, we decodes the interpolated latent representation to the image space. A visual description is shown in Figure 2.

**Properties.** If $K_Z$ is indeed invertible, the empirical mean embedding of samples generated using the proposed algorithm (assuming the preimage is exact) is equal to the empirical mean embedding of the latent representations, indicating a match in distribution. Further, we use locally-weighted Fréchet mean on sphere to construct approximate preimage of sample in RKHS. Since a closed-form solution for the weighted Fréchet mean does not exist on the sphere, we propose to use a simple and efficient algorithm, namely the geodesic interpolation (gI) Salehian et al. (2015), that uses the geodesic on a hypersphere to iteratively computes the weighted Fréchet mean Fréchet (1948) of top $\gamma$ latent representations (see Fig. 4). The algorithm has the following properties: **(i)** the geodesically completeness of the latent space guarantees that the geodesic interpolation is well-defined **(ii)** the geodesic convexity of the latent space guarantees that the output of "gI" algorithm lies on the latent space **(iii)** the "gI" algorithm converges asymptotically to the Fréchet mean Salehian et al. (2015).

> **Geodesic Interpolation (gI)**
>
> (a) Let the inputs be $\{\mathbf{l}_i\}_{i=1}^{\gamma} \subset \mathbf{S}^{d-1}$ and $\{w_i\}_{i=1}^{\gamma}$.
> (b) Initialize $\mathbf{m} = \mathbf{l}_1$.
> (c) For $j = 2, \cdots, \gamma$
>     (a) Set $t = w_j / \sum_{k=1}^{j} w_k$, $\theta = \arccos(\mathbf{m}^\top \mathbf{l}_j)$.
>     (b) Update
> $$\mathbf{m} = \frac{1}{\sin(\theta)} \left( \mathbf{m} \sin\left((1-t)\theta\right) + \mathbf{l}_j \sin\left(t\theta\right) \right).$$
> (d) Return $\mathbf{m}$ as the output.

Figure 4: The geodesic interpolation algorithm

**Choice of Kernels.** In order to be able to linearize the dynamics between the prior and the target distribution, one must first identify a set of nonlinear basis functions such that the corresponding dynamics lies in its span. Known results in dynamical systems guarantee the existence of such a linear operator, such as the Koopman operator Koopman (1931), given an infinite set of basis functions. But rather than identifying certain modes that best characterize the dynamics Williams et al. (2015); Brunton et al. (2016) we care most about minimizing the error of the transferred density, and whether the span of functions is rich/expressive enough and can be efficiently computed. Therefore, the choice of kernel is important since it directly determines the family of functions spanned by the operator. We empirically evaluate the effect of using several different kernels by a simple experiment on MNIST. The MNIST digits are first trained for 100 epochs using an autoencoder with latent space restricted to $S^2$, then samples are generated using procedure described in Figure 3 using the respective kernel function. Subplot (b) and (c) show the generated samples when using Radial Basis Function (RBF) kernel and arc-cosine kernel, respectively. Observe that the choice of kernel has a clear influence on the posterior, but and a kernel with superior empirical behavior would be desirable.

**NTK.** We use Neural Tangent Kernel (NTK) Jacot et al. (2018) as our embedding kernel due to the following properties: **(a)** NTK, in theory, corresponds to a trained infinitely-wide neural network, and can be non-asymptotically approximated Arora et al. (2019). **(b)** For well-conditioned inputs (i.e., no duplicates) on the sphere, the positive-definiteness of NTK is proved in Jacot et al. (2018). Therefore, invertibility of $G_{XX}$ is *guaranteed* if the sampling distribution is restricted on a

> **Generator**
>
> (a) Let $\mathbf{z}^*$ be a sample drawn from $u_{\mathbf{S}^{d-1}}$. We first post-multiply with $k(\mathbf{z}^*, .)$ to get $\widehat{\mathcal{P}}_\epsilon k(\mathbf{z}^*, .) = \Phi_G (G_{XX} + N\epsilon I)^{-1} [k(\mathbf{z}_i, \mathbf{z}^*)]^\top$ (4), where, **(a)** $G_{XX} = [k(\mathbf{z}_i, \mathbf{z}_j)]$. **(b)** $\Phi_G = [k(\mathbf{l}_i, .)]^\top$
>
> (b) Take inner product, $\langle ., . \rangle_{\mathcal{G}}$ of $\Phi_G$ with $\widehat{\mathcal{P}}_\epsilon k(\mathbf{z}^*, .)$ to get $\langle \Phi_G, \widehat{\mathcal{P}}_\epsilon k(\mathbf{z}^*, .) \rangle = \Phi_G^\top \Phi_G (G_{XX} + N\epsilon I)^{-1} [k(\mathbf{z}_i, \mathbf{z}^*)]^\top = [k(\mathbf{l}_n, \mathbf{l}_o)] \cdot ([k(\mathbf{z}_i, \mathbf{z}_j)] + N\epsilon I)^{-1} [k(\mathbf{z}_k, \mathbf{z}^*)]^\top = K_L (K_Z + N\epsilon I)^{-1} \mathbf{v}^\top$, where $K_L = [k(\mathbf{l}_n, \mathbf{l}_o)] \in \mathbf{R}^{N \times N}$ and $K_Z = [k(\mathbf{z}_i, \mathbf{z}_j)] \in \mathbf{R}^{N \times N}$; $\mathbf{v} = [k(\mathbf{z}_k, \mathbf{z}^*)] \in \mathbf{R}^N$.
>
> (c) Notice that both $K_L$ and $(K_Z + N\epsilon I)^{-1}$ can be pre-computed.
>
> (d) Let $\mathbf{s} = \langle \Phi_G, \widehat{\mathcal{P}}_\epsilon k(\mathbf{z}^*, .) \rangle$. Let $\{\mathbf{l}_j\}_{j=1}^{\gamma}$ be the latent representations of $\gamma > 0$ data samples with largest inner products in $\mathbf{s}$ (in descending order).
>
> (e) Return the generated sample $X = D\left( \text{gI}\left( \{\mathbf{l}_j\}_{j=1}^{\gamma}, \{w_j\}_{j=1}^{\gamma} \right) \right)$. Here, $D$ denotes the decoder function, $w_j = \mathbf{s}_j / \sum_i \mathbf{s}_i$, $\text{gI}(., .)$ is the geodesic interpolation algorithm (as presented in Fig. 4).

Figure 3: Step by step details of our sample generation algorithm.

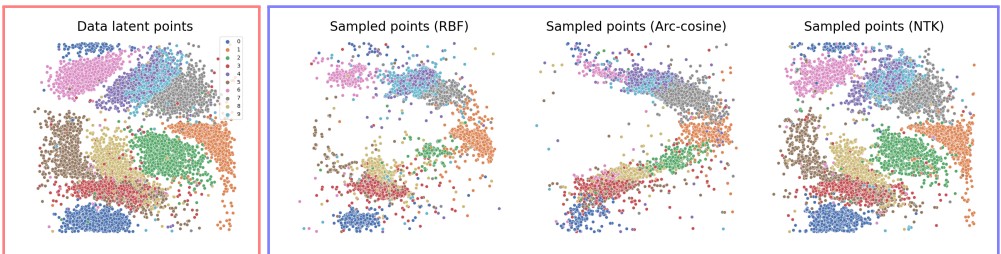

Figure 5: 10k samples from MNIST dataset (*left to right*) (a) projected on $\mathbf{S}^2$ shown in $(\theta, \phi)$ using auto-encoder, and 10K generated samples using (b) RBF (c) arccos (d) NTK. Minimum regularization has been applied to all kernels to ensure invertibility. Color of sampled points represents the class of their 'nearest' point in the feature space.

hypersphere, which is true by our modeling choice. **(c)** Unlike other parametric kernels such as RBF kernels, NTK is less sensitive to hyperparameters, as long as the number of units used is large enough Arora et al. (2019). The embedding of generated samples for MNIST using NTK is also shown in Fig. 5. Observe that the sample distribution more closely resembles the original distribution.

**Nyström's approximation.** A known bottleneck with kernel based methods is instantiating a kernel matrix whose size depends on the dataset size. Here, we use use the Nyström's method Nyström (1930) to obtain approximate kernel matrices. More specifically, a kernel matrix $K \in \mathbf{R}^{N \times N}$ can be approximated by $\widetilde{K} = KS \left(S^T K S\right)^\dagger S^T K$, where $S \in \mathbf{R}^{N \times s}$ is the sampling matrix. To minimize information loss, we use the RecursiveRLS-Nyström algorithm proposed in Musco & Musco (2017) to sample $S$ based on the approximate ridge leverage scores. We refer the readers to the work by Musco & Musco (2017) for detailed algorithm. Notice that after the approximation, we only need to store $KS \in \mathbf{R}^{N \times s}$ and $S^T K S \in \mathbf{R}^{s \times s}$ matrices to obtain the Nyström approximation of $K$, bringing down the memory complexity from $O(N^2)$ to $O(Ns + s^2)$. When $s \ll N$, the savings are significant.

## 4 EXPERIMENTAL RESULTS

We qualitatively evaluate the proposed generative model on two types of applications, **(a)** datasets with a sufficient number of samples to model the data distribution **(b)** datasets with fewer number of samples so that learning the data distribution is harder . In the first case, we use standard vision datasets, including MNIST, CIFAR10, and CelebA, where the number of data samples is larger than the dimension. In order to handle the more challenging second case, we use a dataset of T1 Magnetic Resonance (MR) images from a public brain imaging project called Alzheimer's Disease Neuro-Imaging Initiative (ADNI).

|  | MNIST | CIFAR | CelebA |
|---|---|---|---|
| WGAN | 6.7 | 55.2 | 41.3 |
| WGAN-GP | 20.3 | 55.8 | 30.3 |
| Vanilla VAE | **13.7** | 111.0 | 52.1 |
| Two-stage VAE | 18.3 | 110.3 | 44.7 |
| SRAE$_{rand\ interp}$ | 28.5 | **104.7** | 64.3 |
| SRAE$_{Glow}$ | 26.1 | 110.9 | 50.9 |
| SRAE$_{RBF\text{-}kPF\ 10k}$(*ours*) | 24.7 | 124.5 | 57.7 |
| SRAE$_{NTK\text{-}kPF\ 10k}$(*ours*) | 15.0 | 123.3 | 39.9 |
| SRAE$_{NTK\text{-}kPF\ Nyström}$(*ours*) | 15.9 | 119.4 | **39.2** |

Table 1: Comparative FID values. SRAE indicates an autoencoder with hyperspherical latent space and spectral regularization following Ghosh et al. (2020). Subscripts indicates the corresponding sampling techniques on latent space. *rand_interp*: geodesic interpolation with uniform weights among 10 uniformly sampled latent representations. *Glow*: samples from a Glow model trained on latent representations

The purpose of the first set of experiments is to show that the proposed method yields competitive measures compared to other non-adversarial generative methods while enjoying the benefit of *one step* density estimation. In the second setting, our goal is to demonstrate that unlike traditional non-adversarial models, our proposed method produces reasonable sample generation both quantitative and qualitatively.

**Computer vision datasets:** We evaluate the quality by calculating the Fréchet Inception Distance (FID) Heusel et al. (2017). All models share the same encoder-decoder architecture and trained for 100 epochs. Subscript *10k* indicates the kPF is estimated using 10000 latent points, whereas *Nyström* indicates approximation using 4000 landmark points from all latent points. Comparative results are

shown in Table 1. We compare our results to vanilla VAE Kingma & Welling (2013), 2-Stage VAE Dai & Wipf (2019), and Glow + AE (denoted by $SRAE_{Glow}$ in the table). FIDs of two best GAN methods are also reported from Dai & Wipf (2019) for comparison (note that the reported scores have been optimized by large-scale neural architecture search). We observe that for images with structured feature spaces, e.g., MNIST and CelebA, the proposed method matches or outperforms other non-adversarial generative models. On the contrary, on images with less structured features such as CIFAR10, our model performs worse. We suspect that this is due to a non-smooth latent space, and we evaluate this by showing that visually similar images picked by our operators decodes to out-of-distribution (blurred-out) images (see Fig. 6). Here, we use a regularized AE Ghosh et al. (2020) with a latent space restricted to the hypersphere (denoted by SRAE).

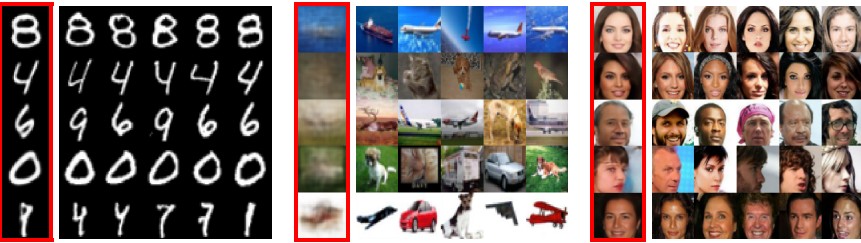

Figure 6: Generations (in red box) and training samples corresponding to the top-5 latent representations used in geodesic interpolation. It can be observed that the samples with top kernel values indeed share high visual similarity.

Further, we present qualitative results on CelebA (in Fig. 7), where we compare our kPF based model with other density estimation techniques on the latent space. Observe that our model generates comparative visual results with $SRAE_{Glow}$. Hence, when a sufficient number of samples are available, our method performs as good as the alternatives, which is beneficial given the efficient training stage. We present training time comparison and show that our proposed method is about $40\times$ faster as in Fig. 9.

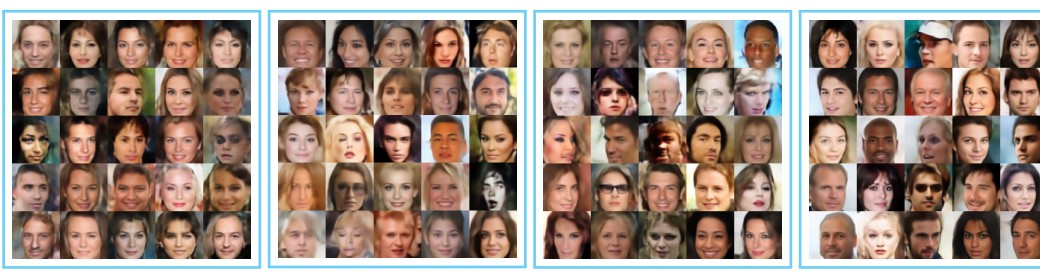

Figure 7: Comparison of different sampling techniques using AE trained on CelebA 64x64. *Left to right: (1) Interpolation among 10 random latent points (2) samples of SRAE+Glow (3) samples of two-stage VAE (4) samples of kPF-flow using 10k latent points*

**Brain Imaging dataset:** In this section, we present results on generating high-resolution ($160 \times 196 \times 160$) 3D brain images from ADNI consists of 183 samples from group AD (diagnosed as Alzheimer's Disease) and 291 samples from group CN (control normals). In this setting (where $n = 474 \ll d = 5017600$), it is often hard to model the data distribution using either variational methods or flow-based methods due to the high variance of the data or the memory-inefficiency of the operations. However, benefiting from the regularized and linear nature of our kernel operator Willoughby (1979); Arora et al. (2020), we can still generate high-quality samples in such resolution that are in-distribution. As before, we present the comparative results with respect to the VAE model. The generated samples presented in Fig. 8 clearly demonstrate that proposed method generates sharper images. To evaluate whether these results are also scientifically meaningful (and not merely visually pleasing), we tested consistency between statistical group difference testing on the real images (groups were AD and CN) and the same testing performed on the generated samples. We performed a FWER corrected two-sample t-test Ashburner & Friston (2000) in a manner consistent with standard practice Ashburner & Friston (2000); Winkler et al. (2014). The results (see Fig 8) show that while there is a deterioration in regions identified to be affected by disease (and different

across groups), most of the statistically-significant regions from the test on the original images are preserved in tests performed on the generated images.

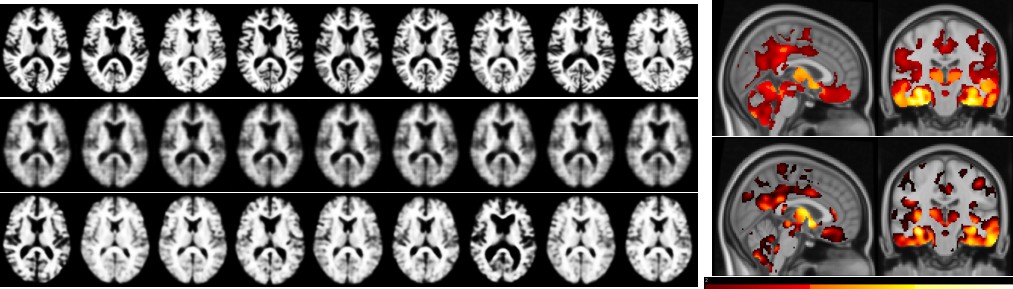

Figure 8: **Left.** *Top: data*, generated samples of *Middle: VAE samples, Bottom: kPF samples*. Standard Glow failed to fit into system memory for image of this resolution. **Right.** Statistically significant regions *Top: data, Bottom: samples* are shown in negative log $p$-value thresholded at $p < 0.01$.

*Take-home message:* We conclude that due to the kernel approach and one-step formulation of the proposed model, it offers better memory and sample efficiency than other non-adversarial generative methods. This is useful in many tasks, e.g., medical image analysis, where the number of samples is usually much less than the dimension of data. Moreover, in case of datasets with larger number of samples, the proposed method still performs well (and sometimes better) compared to alternatives, with a sizably smaller resource footprint.

## 5 DISCUSSION

We have shown that when a dataset allows a low-dimensional structured latent space representation, a kernel Perron-Frobenius operator can provide an efficient strategy to simplify flow-based generative models to transfer the density from a known simple distribution to the empirical data distribution. While the algorithmic simplifications proposed here can be variously useful, deriving the density of the posterior given a mean embedding or providing an exact preimage for the generated sample in RKHS remain unresolved at this time. While the problem of deriving the density has been addressed in Schuster et al. (2020), identifying a pre-image can be hard and often ill-posed. We also note that our geodesic interpolation only converges asymptotically to the weighted Fréchet mean which we use as the approximate preimage, and additional improvements on this front are possible.

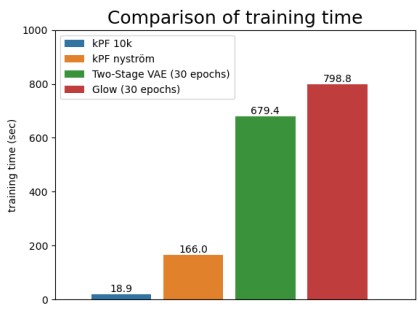

Figure 9: Comparision of additional training time to other density estimation model on the latent space.

## 6 CONCLUSION

In this paper, we show that with the assistance of recent developments in regularized autoencoder and neural kernels, a linear kernel transfer operator can potentially be an efficient substitute to flow-based generative models. Our proposed method shows comparable empirical results to other state-of-the-art generative models on several computer vision datasets, while enjoying higher computational efficiency. The results on brain imaging data also shed light on the potential application on high-volume data generation, which are typically hard to model using existing methods.

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

# A APPENDIX

## A.1 PROOF FOR THEOREM 1

The proof follows easily from the fact that $\hat{\mathcal{P}}_{\mathcal{E}}$ is linear and independent to $\{x_i\}_{i=1}^n$,

$$E[\hat{\mu}_{t+\tau}^*] = E[\frac{1}{n}\sum_{i=1}^n \hat{\mathcal{P}}_{\mathcal{E}}\phi(x_i)] = E[\hat{\mathcal{P}}_{\mathcal{E}}]E[\frac{1}{n}\sum_{i=1}^n \phi(x_i)] = \mathcal{P}_{\mathcal{E}}\mu_t = \mu_{t+\tau}$$

## A.2 ABLATION STUDIES

We present ablation studies on MNIST (Fig. 10), showing importance and different components of our proposed method. Our proposed framework in general consists of **(a)** learning latent space **(b)** estimating transfer operator using kPF. We replace the first part in two ways **(i)** remove AE and directly transfer the distribution from image space to the known distribution **(ii)** replace AE with known invertible basis representation, e.g., mapping onto Fourier basis . In both these cases, we observe that the generation quality deteriorates. Notice that such degradation can be justified due to the lack of smoothness on the input space. Next, we fixed the AE and replace the kPF operator with a normalizing flow model Kingma & Dhariwal (2018) as transfer operator. Observe that the generation quality are comparable between the two methods.

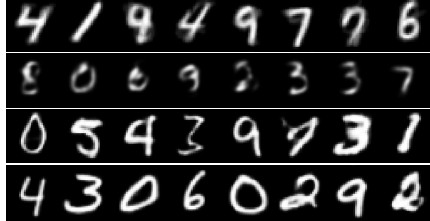

Figure 10: *Top to bottom: no latent space with kPF, (2) interpolation in Fourier space, (3) Glow as transfer operator, (4) kPF as transfer operator*

We further show the advantage of using a regularized AE over a regular one. The regularization scheme we applied is derived from Ghosh et al. (2020), where we apply spectral normalization on all encoder-decoder transformations. Regularization has been shown to lead to smoother latent spaces Alain & Bengio (2014),

|  | MNIST | CIFAR | CelebA |
|---|---|---|---|
| SAE$_{NTK\text{-}kPF\ 10k}$ | 22.3 | 133.6 | 41.2 |
| SRAE$_{NTK\text{-}kPF\ 10k}$ | 15.0 | 123.3 | 39.9 |

Table 2: vanilla vs. regularized AE.

and therefore is critical in our interpolation-based sampling approach. The effect of such regularization can be seen in Table 2, where it leads to higher FIDs in all three datasets using smoother latent space.

*Take-home message:* To summarize, the presented ablation study demonstrates the importance of the latent space as well as the richness of our one-step estimated operator. Moreover, the boundedness and smoothness of the latent space is crucial as can be justified by replacing AE with a Fourier basis or by using a regularized AE.

## A.3 ADNI DATASET AND BRAIN IMAGE PROCESSING

Data used in preparation of this article were obtained from the Alzheimer's Disease Neuroimaging Initiative (ADNI) database (adni.loni.usc.edu). As such, the investigators within the ADNI contributed to the design and implementation of ADNI and/or provided data but did not participate in analysis or writing of this report. A complete listing of ADNI investigators can be found at: .

The T1 MR brain data we use consists of images from 184 subjects diagnosed with Alzheimers's disease and 292 control normal subjects. Images were first coregisted to a MNI template and segmented to keep only the white matter and grey matter. Then, all images were resliced and resized to $160 \times 196 \times 160$ and rescaled to the range of $[-1, 1]$. Voxel-based morphometry (VBM) was used to obtain the $p$-value map of data and generated images.

## A.4 FID SCORES ON CELEBA DATASET USING RAE DECODER

|     | WGAN | 2-Stage VAE | SAE$_{kPF}$ | SRAE$_{kPF}$ |
|-----|------|-------------|-------------|--------------|
| FID | 56.5 | 45.25       | 46.1        | **44.6**     |

Table 3: FID scores using the architecture and training protocol from Ghosh et al. (2020)

## A.5 IMAGE QUALITY EVALUATION

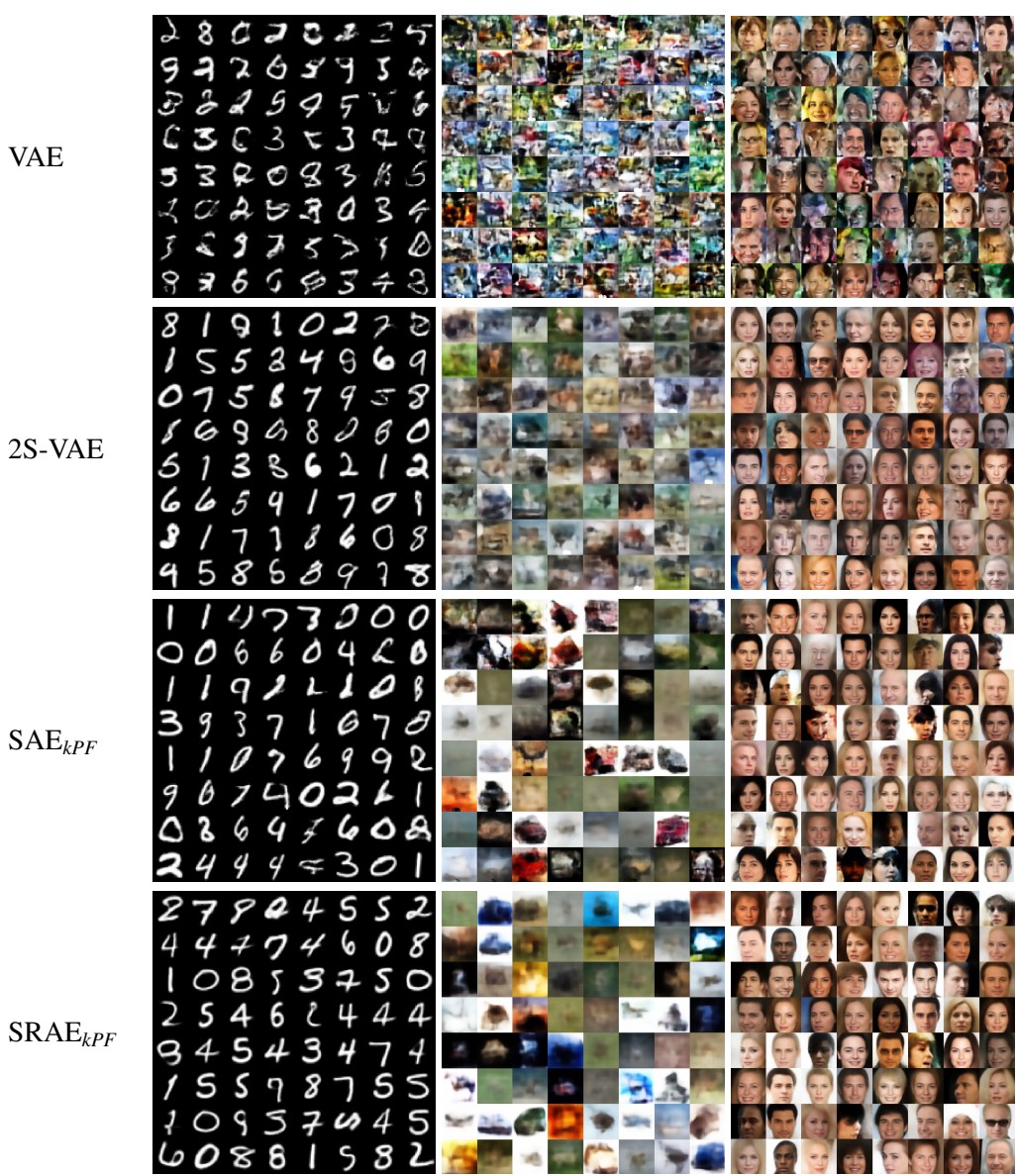

Figure 11: Randomly picked samples from (1) VAE (2) 2-Stage VAE (3) SAE$_{NTK\text{-}kPF\ 10k}$ (4) SRAE$_{NTK\text{-}kPF\ 10k}$. All models trained with the architecture in Ghosh et al. (2020) without tuning for hyperparamters

## A.6 CHOICE OF $\gamma$

In the inference stage, our proposed method finds the approximate preimage of the transferred kernel embeddings by interpolating among the top $\gamma$ latent representations of the training samples weighted by their kernel values. The choice of $\gamma$ therefore has implications on the generation quality. From Figure 12, we can observe that, in general, FID worsens as $\gamma$ increases. This observation aligns with our intuition of preserving only the local similarities in kernel embeddings, and similar idea has

been used in the literature Kwok & Tsang (2004). However, significantly decreasing $\gamma$ leads to the undesirable result where the generator merely generates the training samples (in the extreme case where $\gamma = 1$, generated samples will just be reconstructions of training samples). Therefore, in our experiments, we choose $\gamma = 10$ to achieve a balance between generation quality and the distance to training samples.

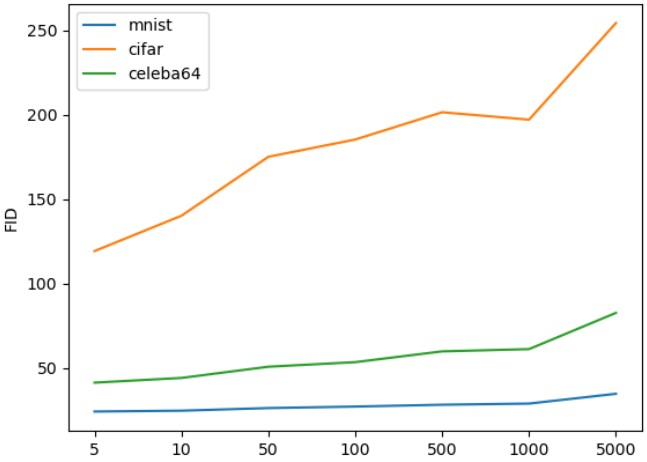

Figure 12: FID versus choice of $\gamma$

## A.7 Experimental specifications

For all experiments in Table 1 except the GAN experiments, we adapted a custom ResNet implementation where each block has residual connection

$$\rightarrow \text{BatchNorm} \rightarrow \text{Swish} \rightarrow \text{3x3 Conv} \rightarrow \text{BatchNorm} \rightarrow \text{Swish} \rightarrow \text{3x3 Conv} \rightarrow$$

The detailed architecture is given by the following table

|  | MNIST | CIFAR-10 | CelebA |
|---|---|---|---|
|  | 5x5 conv, stride 1 | | |
| Encoder | ResBlock$_{32}$ × 2
ResBlock$_{64}$ × 2
ResBlock$_{128}$ × 2
ResBlock$_{256}$ × 2 | ResBlock$_{32}$ × 2
ResBlock$_{64}$ × 2
ResBlock$_{128}$ × 2
ResBlock$_{256}$ × 2 | ResBlock$_{32}$ × 2
ResBlock$_{64}$ × 2
ResBlock$_{128}$ × 2
ResBlock$_{256}$ × 2
ResBlock$_{512}$ × 2 |
| Decoder | ResBlock$_{256}$ × 2
ResBlock$_{128}$ × 2
ResBlock$_{64}$ × 2
ResBlock$_{32}$ × 2 | ResBlock$_{256}$ × 2
ResBlock$_{128}$ × 2
ResBlock$_{64}$ × 2
ResBlock$_{32}$ × 2 | ResBlock$_{512}$ × 2
ResBlock$_{256}$ × 2
ResBlock$_{128}$ × 2
ResBlock$_{64}$ × 2
ResBlock$_{32}$ × 2 |
|  | 5x5 conv, stride 1 | | |

Table 4: Detailed network architecture for experiments in Table 1. Subscript denotes the number of input channels. Upsampling and downsampling are performed using strided convolutions.

We train each model for 100 epochs on every dataset using Adam optimizer Kingma & Ba (2014) with $\beta = (0.9, 0.999)$ and batch size of 512. The learning rate starts from 1e-3 and is halved every 30 epochs. We used implemetation of Novak et al. (2020) to estimate the NTK with four fully-connected layers, each with 10000 units, and Erf activation. VAE and 2-Stage VAE are trained using learnable-$\gamma$ as in Dai & Wipf (2019). For experiments in A.4 and A.5, we adopted the training procedure and architecture in Ghosh et al. (2020)

