# OpenReview forum: "Can Kernel Transfer Operators Help Flow based Generative Models?"
_ICLR.cc/2021/Conference — Reject_

### Official Review · AnonReviewer4 · 2020-10-13
**Very confusing paper.**

**Rating:** 2
**Confidence:** 4

**Review:**

This paper proposes a method for a deep generative model utilizing a trained autoencoder in conjunction with kernel conditional embeddings. The authors validate their model experimentally on CelebA and a brain imaging dataset.

Overview : I found this paper very confusing with crucial mathematical definitions and exposition often inconsistent or imprecise. Outside of this the general exposition in many ways very strange and confusing. Finally the lack of experimental code makes this paper a very clear reject in my opinion. The rest of the review will list some concrete examples of the issues I mentioned.

Mathematical imprecision:
- The Phi operator in this paper is problematic. I'm guessing this is meant to be the kernel feature map, but in Def 1 it is introduced as an arbitrary element of the RKHS. In Def 5 Phi_H and Phi_G are initially introduced via the phrase "where an element is denoted by Phi_G" seeming to imply that is again simply an example element of their respective RKHSs. Later in this definition the well-known (kernel) cross covariance operator "C_YX = E_YX [ Phi_G (y) \otimes Phi_H (x)]" which I am assuming implies that it is the kernel feature map, since otherwise this quantity is simply a scalar. Finally in the description of their main algorithm step 1 the refer to Phi_G^T which is highly problematic since Phi_G is presumably well defined here, meaning it is the kernel feature map which is not a linear operator (the whole point of kernel methods being that this operator is nonlinear), thus its "transpose" isn't something that makes sense. The authors give a vector which is equal to the expression in which Phi_G appears, this however depends on a \bold{1} vector defined as " are the top lambda (a hyper-parameter) latent representations" which is not explained further. Presumably this is referring to approximate inverse kernel feature map (which has seen some research), but this needs to be further explained, or at the very least a citation is needed.

General Exposition:
-I am not sure why the approach to the paper is being framed in a Markov setting, when it seems like the authors really only need kernel conditional embeddings which have the exact same form as the Perron Frobenius operator the authors use. The authors even cite the kernel conditional embedding paper Song et al. 2009. Things like (1) which only serve to complicate the exposition unnecessarily considering there is no Markov process type of structure in the main algorithm.

-p. 2 Rationale: "For instance, it is known that given infinite-dimensional observables of the input measurements, there exists a linear transfer operator that perfectly pushes one probability density to another. Of course, this is not practically beneficial because explicitly constructing such infinite-dimensional observables for the data could be intractable. Nonetheless, results in control theory demonstrate that the idea can still be effective in specific cases, using approximations with either spectral analysis of large but finite number of observable functions " This sort of thing needs to be decompressed, I don't know what is meant by an "observable" I assume it has something to do with control theory or Markov process theory. This seems to niche for me for the deep learning crowd.

-The Riemannian geometry in the paper needs to be explained more for the average deep learning researcher, e.g. its not clear why the "geodesic interpolation" is doing at or why its algorithm makes sense.

-The main algorithm needs to be explained with more details as to _why_ we are doing the steps.

---

> ### Author Response · Authors · 2020-11-19
> **Response to the reviewer**
>
> We are thankful for your review and appreciate for pointing out the clarity issues with our work. We have revised the draft significantly and hope to clarify your confusion.
>
> **I found this paper very confusing with crucial mathematical definitions and exposition often inconsistent or imprecise. Outside of this the general exposition is in many ways very strange and confusing. Finally the lack of experimental code makes this paper a very clear reject in my opinion. The rest of the review will list some concrete examples of the issues I mentioned.**
>
> Ans: We made a significant revision of the manuscript and kindly request the reviewer to read the revised draft. We hope the revised form will clarify the confusion. We have also uploaded a jupyter notebook example for the reviewer to experiment with, as well as giving a detailed specification of our experimental setups for reproducibility.
>
> **The Phi operator in this paper is problematic. I'm guessing this is meant to be the kernel feature map, but in Def 1 it is introduced as an arbitrary element of the RKHS. In Def 5 Phi_H and Phi_G are initially introduced via the phrase "where an element is denoted by Phi_G" seeming to imply that is again simply an example element of their respective RKHSs. Later in this definition the well-known (kernel) cross covariance operator "C_YX = E_YX [ Phi_G (y) \otimes Phi_H (x)]" which I am assuming implies that it is the kernel feature map, since otherwise this quantity is simply a scalar. Finally in the description of their main algorithm step 1 the refer to Phi_G^T which is highly problematic since Phi_G is presumably well defined here, meaning it is the kernel feature map which is not a linear operator (the whole point of kernel methods being that this operator is nonlinear), thus its "transpose" isn't something that makes sense. The authors give a vector which is equal to the expression in which Phi_G appears, this however depends on a \bold{1} vector defined as " are the top lambda (a hyper-parameter) latent representations" which is not explained further. Presumably this is referring to an approximate inverse kernel feature map (which has seen some research), but this needs to be further explained, or at the very least a citation is needed.**
>
> Ans: It is a typo by us. $\Phi_H$ and $\Phi_G$ in Def 5 should be lowercase instead, indicating the feature mappings. We apologize for the confusion and they have been updated in the text.
>
> **I am not sure why the approach to the paper is being framed in a Markov setting, when it seems like the authors really only need kernel conditional embeddings which have the exact same form as the Perron Frobenius operator the authors use. The authors even cite the kernel conditional embedding paper Song et al. 2009. Things like (1) which only serve to complicate the exposition unnecessarily considering there is no Markov process type of structure in the main algorithm.**
>
> Ans: Yes, this is correct. We have significantly updated the description which has hopefully simplified the presentation.
>
> **p-2 Rationale: "For instance, it is known that given infinite-dimensional observables of the input measurements, there exists a linear transfer operator that perfectly pushes one probability density to another. Of course, this is not practically beneficial because explicitly constructing such infinite-dimensional observables for the data could be intractable. Nonetheless, results in control theory demonstrate that the idea can still be effective in specific cases, using approximations with either spectral analysis of large but finite number of observable functions " This sort of thing needs to be decompressed, I don't know what is meant by an "observable" I assume it has something to do with control theory or Markov process theory. This seems to niche for me for the deep learning crowd.**
>
> Ans: We apologize for the confusion. We have rewritten this description by restating the intuition described in the literature. We believe that this should clarify the intuition and the ideas much better.
>
>
> **The Riemannian geometry in the paper needs to be explained more for the average deep learning researcher, e.g. its not clear why the "geodesic interpolation" is doing at or why its algorithm makes sense.**
>
> Ans: The geodesic interpolation is an efficient way to approximate the weighted Fréchet mean, and it is used in our algorithm to construct an approximate preimage of the generated feature map (so that it can be decoded to the image space). Constructing the preimage is not the main focus of our generative procedure and there are potentially better alternatives, e.g. MDS. We have incorporated more detailed reasoning of the geodesic interpolation in Section 3.2.
>
> **The main algorithm needs to be explained with more details as to why we are doing the steps.**
>
> Ans: We have updated a more detailed description of the main algorithm and some clarification in the previous section.

---

### Official Review · AnonReviewer1 · 2020-10-29

**Rating:** 5
**Confidence:** 4

**Review:**

##### Post-rebuttal update

Dear authors, Thanks for the response. I strongly encourage you to revise the paper using languages other than flow-based generative models.
From the rebuttal I could not see whether you understand my point or not: The conditional mean embedding operator defines an **integration** instead of an invertible **transformation**, which differentiate itself from any flow-based model (at least those you referenced).
As for now, I cannot recommend for acceptance.

--------------

The paper proposes a different kind of generative model that is composed of autoencoder in the bottom, and a standard distribution p(z), and a conditional kernel mean embedding defined by a collection of sample pairs (z_i, l_i). The distribution of the latent codes p(l) is modeled by the kernel sum rule that corresponds to the marginalization \int p(l|z)p(z) dz (ultimately defined by the collection of sample pairs).

##### Originality & Significance
The proposal of modeling the latent distribution of an autoencoder in a nonparametric way (using kernel mean embeddings) is original to my best knowledge. To do this, you need to solve the pre-image problem (we can get the mean embedding of p(l) through kernel operations but it is generally difficult to map it back to samples). This paper adopts a previous approach (geodesic interpolation, Salehian et al. 2015), which I'm not aware of but appears to be theoretically sound: "the gI algorithm converges asymptotically to the true solution."

##### Clarity
The clarity is low. The title (and introduction) is very misleading as it says a lot about flow-based generative model. However, this paper is really about modeling the latent distribution of an autoencoder using conditional mean embeddings (what the paper calls embedded Perron-Frobenius operator).. In every aspect I find it more close to a latent-variable model (there is a latent prior over the z space). And the distribution in the L space is the result of the kernel sum rule that corresponds  to the integral \int p(l|z)p(z) dz. Why do the authors prefer to call it a flow?

##### Others

* One strength of this paper is that the work is very careful in choosing appropriate kernels, the importance of which is often ignored in works that deploy kernel methods. Figure 5 clearly shows the benefits of a good kernel (NTK) for this task. The result also shows that Nystrom methods seem to work well with NTK

* Definition 1: the RKHS should be the completion of the span.

* I don't understand the method used to generate figure 6 (1), what is "interpolation among 10 random latent points"? Why is this a sensible baseline?

---

> ### Author Response · Authors · 2020-11-19
> **Response to the reviewer**
>
> We truly appreciate your kind review and hope the following explanations and revisions will address your concerns about the paper.
>
> **The clarity is low. The title (and introduction) is very misleading as it says a lot about flow-based generative model. However, this paper is really about modeling the latent distribution of an autoencoder using conditional mean embeddings (what the paper calls embedded Perron-Frobenius operator). Why do the authors prefer to call it a flow?**
>
> Ans: On page 2 as well as in the common response, we have provided additional details on why we decided to set up the algorithm as a simplification to the transformation mapping learned in flow-based generative models. We agree with the reviewer although the choice of where the idea is deployed (and what motivated the simplifications) influenced the title and the introduction.
>
> **One strength of this paper is that the work is very careful in choosing appropriate kernels, the importance of which is often ignored in works that deploy kernel methods. Figure 5 clearly shows the benefits of a good kernel (NTK) for this task. The result also shows that Nystrom methods seem to work well with NTK**
>
> Ans: Thank you for appreciating our work. Indeed, the choice of kernel seems to have significant influence on the accuracy of the fitted distribution as well as the sample quality. During our experiment, we also observe that the type of activation function used to estimate NTK has an influence on the generation quality, and we are glad to provide more details if requested.
>
> **Definition 1: the RKHS should be the completion of the span.**
>
> Ans: Thank you for the comment. We have updated the definition in the revised text.
>
> **I don't understand the method used to generate figure 6 (1), what is "interpolation among 10 random latent points"? Why is this a sensible baseline?**
>
> Ans: The purpose of this demonstration is to show why the choice of interpolants is important. In figure 6(1), we showed that by interpolating among random ten latent points, the quality of samples does not compete with the samples generated using our procedure, which uses the same number of latent point to interpolate. In terms of FID score, the randomly interpolated samples are significantly worse on MNIST and CelebA, but surprisingly outperforms all other non-adversarial methods, which has been similarly reported in Ghosh et al. (2020). Moreover, as the output of the random interpolation is reasonable, this also implies the smoothness of the latent space.
>
> Reference:
> Partha Ghosh, Mehdi S. M. Sajjadi, Antonio Vergari, Michael Black, and Bernhard Scholkopf. From variational to deterministic autoencoders.  In International Conference on Learning Representations, 2020. URL https://openreview.net/forum?id=S1g7tpEYDS.

---

### Official Review · AnonReviewer2 · 2020-10-30
**Review for Can Kernel Transfer Operators Help Flow based Generative Models?**

**Rating:** 5
**Confidence:** 3

**Review:**

This paper starts with an autoencoder trained on vision data. Autoencoders aren't generative models, strictly, so in order to make it so, they leverage a simple linear transformation on over RKHSs. The kernel they use is the NTK. In order to generate, the construct a reduced sample version of the Kernel using Nystrom's approximation, then use a geodesic interpolation algorithm to inverse map the transferred sample from the prior in the RKHS space back to the latent space for use in the decoder.

The method appears to work, which in itself is very interesting as this means one should be able to construct generative models (a la VAEs) starting only from a simple autoencoder. The results aren't that impressive compared to the baselines compared, but surprisingly they don't compare to any flow-based algorithms, such as Normalizing Flows or related methods like Hamiltonian Variational Inference. VAEs using NICE or Real-NVP to transfer the latent space from a autoencoder would be good baselines.

It would be good to have clearer references for the "two-stage VAE" in the experimental results section: I was unable to easily find what this model is from this draft.

I found the geodesic interpolation part a bit difficult to follow and not clearly motivated. Could you go into more detail on its component parts and why they were included (e.g., what appears to be the importance sampling part)? Linking this more clearly to the "properties" section would help I think.

What about the effect of the pretrained AE? Is reconstruction a good measure for the fitness of an AE for this or is the regularization important? How does one know one has a good autoencoder for this procedure? Could you provide results comparing metrics on the autoencoder (e.g., reconstruction loss, L2 or L1 on the parameters or latent space) vs NLL using the kernel method?

Some other comments:
Section 3.2 seems to have a few errors:
on 1. k -> k_z
on 2. is lambda supposed to be gamma?

---

> ### Author Response · Authors · 2020-11-19
> **Response to the reviewer**
>
> We sincerely thank you for your kind review and truly appreciate that you find our work interesting. We hope these revisions we made based on your suggestions help resolve your concerns.
>
> **Surprisingly they don't compare to any flow-based algorithms, such as Normalizing Flows or related methods like Hamiltonian Variational Inference. VAEs using NICE or Real-NVP to transfer the latent space from an autoencoder would be good baselines.**
>
> Ans: We appreciate the comment. Actually, the experiment with the model labeled SRAE_{glow} is indeed the one which transfers the latent space from an autoencoder using flow-based generative model. In this revision, we have clarified this point.
>
> **It would be good to have clearer references for the "two-stage VAE" in the experimental results section: I was unable to easily find what this model is from this draft.**
>
> Ans: We have integrated the reference in the main text. Thank you for the suggestion.
>
> **I found the geodesic interpolation part a bit difficult to follow and not clearly motivated. Could you go into more detail on its component parts and why they were included (e.g., what appears to be the importance sampling part)? Linking this more clearly to the "properties" section would help I think.**
>
> Ans: In the literature Honiene et al. (2011), the authors used a weighted linear interpolation to solve the pre-image problem. This can be viewed as a simple heuristic for the preimage problem. But notice that in our setup the latent space is spherical, hence a simple linear interpolation will not result in a point on the hypersphere. To avoid this problem, we simply resort to geodesic interpolation. Unfortunately, geodesic interpolation does not have a closed-form solution on the hypersphere unlike Euclidean spaces. Consequently, we use the recursive form of weighted Frechet mean, proposed in Salehian et al., directly although other strategies can easily be plugged in.
> Hence, gI (geodesic interpolation) is the recursive computation of Frechet mean. This algorithm is provably convergent and has a linear convergence rate (as shown in Salehian et al.).
>
> **What about the effect of the pre-trained AE? Is reconstruction a good measure for the fitness of an AE for this or is the regularization important? How does one know one has a good autoencoder for this procedure? Could you provide results comparing metrics on the autoencoder (e.g., reconstruction loss, L2 or L1 on the parameters or latent space) vs NLL using the kernel method?**
>
> Ans: (1) Unfortunately we are not able to utilize an existing pre-trained AE since in our method the latent representations are required to be supported by sphere. (2) Apart from the requirement for hyperspherical latent space, the autoencoder in our proposed method is identical to a conventional AE, and therefore a balance between fitness and regularization is important. In our experiments, we do not observe a significant drop in reconstruction metrics when regularization is applied (spectral norm in each layer in our case). (3) Due to the time limit, we have not been able to compare the effect of different types of loss on the kernel method. For simple cases on MNIST, we encourage you to try with the provided jupyter notebook example. For other experiments, we plan to update this result shortly in the text if time allows.

---

### Official Review · AnonReviewer3 · 2020-11-03
**Interesting concept but lack of clarity in prior work and explanation of the model + non-reproducible experiments**

**Rating:** 5
**Confidence:** 4

**Review:**

**** Summary ****

The authors build a generator that builds on top of the latent space of a “well-trained auto-encoder”. The generator consists of several steps: 1) sampling an latent element from the spherical latent space, 2) using a kernel Perron-Frobenius operator to embed the sampled latent element 3) selecting latent representations of real samples based on the indices of the highest values of the newly calculated embedding 4) calculating the geodesic interpolation of these latent representations 5) using the decoder of the autoencoder to generate new samples from the latent element resulting from geodesic interpolation. The authors describe the related work, the methods they propose and present experimental results on 4 datasets.

To summarise, I believe that the contributions of the authors are the following:
- Proposing a generator that relies on a kernel Perron-Frobenius operator (relevant: medium)
- Comparing variants of their generator to state of the art generative models on four datasets (relevance: medium-low)

**** Positive points of the paper: ****

Relation to prior work and clarity:
The authors attempt to describe extensively the theory behind their work. However, as we will see below, the prior work section could be clearer. Additionally, the link between concepts introduced in prior work and presented in the newly developed algorithm should be more straightforward. See below for detailed comments on improvable points.

Novelty:
The method consists of a novel combination between an existing generative model and an existing kernel transfer operator. I believe that the combination is new however it is not thoroughly justified.

Theoretical setting:
The authors introduce in prior work several of the necessary definitions for the following part. However, some required hypotheses do not always seem to be carefully verified. See below for detailed comments on improvable points.

Experimental setting:
The authors compare their model to several state-of-the-art generative models and show some results on one dataset using different types of kernels. However, details from the experiments are missing to be able to evaluate it thoroughly and the experiments are not reproducible. See below for detailed comments on improvable points.

**** Improvable points of the paper ****

Relation to prior work and clarity:
In general, some sentences of the paper would benefit from being rewritten as they are too convoluted in my opinion. For example, p3, Section 3, first paragraph: “We will follow this rational… variance is decreased”; p6, second paragraph, “to push a uniform … wrapped on S2”. Other sentences are not theoretically justified: p3, last paragraph: “Moreover, the projection … that we will use later”; p.4 Remark 2, “But notice that performing these …. is more sensible”.

Additionally, it seems that the prior work section starts with the introduction and ends at Section 3.2 (excluded). The structure of the paper does not allow a simple identification of the prior work from the method as Section 3 contains prior work and new developments.

Also, it seems that most of the elements of the prior work sections are coming from Fukumizu et al. (2013) and Klus et al. (2020). However, it is not always possible to understand the content of the paper without having to read the definitions/properties in the respective papers. For example, a) Fukumizu et al does not seem to mention the name of the Perron-Frobenius operator, b) in Definition 5 it is not clear why there is a P_eps and a P_k, especially when the variable eps is not defined before, c) Definition 5, it is not clear whether P_kg should belong or not to an RKHS for the Definition to be valid (as mentioned in Klus et al.) d) p4, last paragraph, the equation giving P_kg and P_eps are not introduced properly (especially the need for epsilon) and we need to look at the referred paper to understand where it is coming from e) p5, second paragraph, G_xx is not defined (I understand it is the gram matrix but it should be written), the same holds for N f) p5, second paragraph, equation 3 should be better explained and a reference to Klus et al. and Fukumizu should be made. g) p5, second paragraph, we don’t understand what mu_t is. h) Proposition 1 is not proven (and is written as a definition for reversibility in Klus et al) i) p5, in Notations, \mathcal{L} is not introduced. j) p5, Section 3.2, D (decoder, but should be introduced) and top \lambda are not clear. k) p5, Section 3.2, it is not clear what s is and it has a different typesetting in the bullet points 1 and 2. l) The transpose symbol is not always the same along the paper and can be confused with the iteration number in Figure 4. m) it is not clear where Proposition 1 is needed in practice.

Theoretical setting:
Some concepts are not well defined. For example:
- The authors always talk about a well-behaved VAE. However, the authors do not define what they mean by well-behaved. In addition, looking at Table 1, it seems that the proposed model is not systematically the best performing when the VAE is well behaved (the FID for vanilla VAEs are often better than the proposed model).
- I am not sure what “true solution” means p6, end of Properties. It would be useful if the authors could give the intuition to make the reader understand what the true solution means and the need of the geodesic interpolation.
- Some hypotheses are not verified, for example, a) is K_z invertible in practice (see Properties p6), b) is the experimental covariance is a good approximation for the true covariance.

In general, the algorithm seems to be explained twice with different degrees of precision at the bottom of p5 and in Figure 3 (which should be referred to in the text). The paper would benefit from merging both explanations.

It is not clear to me why the new sampling strategy would follow better the distribution of real samples. I think this should be better explained.

Experimental setting:
The experiments are described in a way that is not reproducible. We do not know which hyperparameters are chosen (for the AE and for the new development) and how they were chosen. We also do not understand how the 4000 landmark points are selected. It is not clear what SRAE_rand_interp is. I do not understand how Figure 5 is done, is it 2D projection of the generated samples? If yes, with which dimension reduction method? (“wrapped on S^2” a bit above the figure is not clear).

A better ablation study could be done:
- In the Supplementary, the authors say that they use a regularised AE (with spectral normalisation) for the SRAE. However, the results of a regularised AE without the development of the authors are not shown.
- The authors do not justify theoretically the need for selecting the top \gamma samples. Therefore, it would be interesting to vary \gamma and observe differences in the results.

In general, the results seem rather poor. The new method outperforms vanilla VAE or basic SRAE only in one of the three datasets for which we have quantitative results. The generative adversarial models outperform systematically the proposed model.


**** Typos: ****
“Which is benefit, given the efficient” -> beneficial, efficiency
Grammian -> Gramian or Gram

*****************
RECOMMANDATIONS:
Because of the points enumerated above, I recommend to reject the paper in its current form. I would increase the grade if a) the paper is rewritten in order to be more structured and clear b) the experiment section is better described and more thorough (ablation study) c) some answers are brought to the theoretical concerns.


*****************
AFTER REBUTTAL
I read the rebuttal of the authors and the other reviews. I will increase the grade to a 5.

The reasons are that I think that the authors improved significantly the paper with this revision. However, I believe that the paper would benefit from experiments on a larger number of datasets, in order to better understand on which type of datasets their method shows better performance.

---

> ### Author Response · Authors · 2020-11-19
> **Response to the reviewer (part 1/3)**
>
> We truly appreciate your thorough and thoughtful review, and we have revised our paper accordingly. We hope the clarifications we made will help address your concerns.
>
> **1. Relation to prior work and clarity: In general, some sentences of the paper would benefit from being rewritten as they are too convoluted in my opinion. For example,**
> **1) p3, Section 3, first paragraph: “We will follow this rational… variance is decreased”**
>
> Ans: We have removed the redundant phrasing in the revised version.
>
> **2) p6, second paragraph, “to push a uniform … wrapped on S2”.**
>
> Ans: We have provided a clearer description for the MNIST experiment on S^2.
>
> **Other sentences are not theoretically justified:**
>
> **3) p3, last paragraph: “Moreover, the projection … that we will use later”;**
>
> Ans: The necessity of restricting the latent space to hypersphere is linked to the known invertibility guarantees of NTK on sphere supported data, which is discussed later in the NTK part of section 3.2. Based on this comment, we have provided clearer reference to the discussion.
>
> **4) p.4 Remark 2, “But notice that performing these …. is more sensible”.**
>
> Ans: In general, the dynamics to the target/template may not have a linear solution in the input space, but may exist in another high-dimensional space spanned by non-linear maps of the inputs, such as an RKHS. The phrasing was a bit confusing in the original version. We hope that in the revision it makes more sense.
>
> **5) The structure of the paper does not allow a simple identification of the prior work from the method as Section 3 contains prior work and new developments.**
>
> Ans: Based on this suggestion, we have clearly delineated the prior work on efficient flow-based generative models on page 2.
>
> **2. Also, it seems that most of the elements of the prior work sections are coming from Fukumizu et al. (2013) and Klus et al. (2020). However, it is not always possible to understand the content of the paper without having to read the definitions/properties in the respective papers. For example,**
> **1) Fukumizu et al does not seem to mention the name of the Perron-Frobenius operator**
>
> Ans: This citation appeared in the wrong place, we have fixed it in the revision. The PF operator is from Klus et al (2020).
>
> **2) in Definition 5 it is not clear why there is a $P_eps$ and a $P_k$, especially when the variable eps is not defined before,
> 3) Definition 5, it is not clear whether $P_kg$ should belong or not to an RKHS for the Definition to be valid (as mentioned in Klus et al.)
> 4) p4, last paragraph, the equation giving $P_kg$ and $P_eps$ are not introduced properly (especially the need for epsilon) and we need to look at the referred paper to understand where it is coming from**
>
> Ans: (Answer to 2.2-2.4) We have reworked the text around Def 5. The notion of P_k is now only briefly mentioned to reduce confusion when introducing work of Klus et al. (2020), since we mainly work with the embedded Perron-Frobenius operator P_eps. The mean embedding operator eps is now introduced in the preliminary. We have also updated the discussion of the need for epsilon before Def 5.
>
> **5) p5, second paragraph, G_xx is not defined (I understand it is the gram matrix but it should be written), the same holds for N**
>
> Ans: We have fixed the definition of these notations.
>
> **6) p5, second paragraph, equation 3 should be better explained and a reference to Klus et al. and Fukumizu should be made.**
>
> Ans: Yes, we fixed this in the revision.
>
> **7) p5, second paragraph, we don’t understand what mu_t is.**
>
> Ans: We have added text to explain mu_t when we are using it at the bottom of p.4 and in Theorem 1.
>
> **8) Proposition 1 is not proven (and is written as a definition for reversibility in Klus et al)**
>
> Ans: Proposition 1 is proved as Proposition 3.7 in Klus et al. We have clarified in the text.
>
> **9) p5, in Notations, \mathcal{L} is not introduced.**
>
> Ans: We have removed this notation of \mathcal{L}, and the definition of \mathcal{L} is given on page 3 when the autoencoder is introduced.
>
> **10) p5, Section 3.2, D (decoder, but should be introduced) and top \lambda are not clear.**
>
> Ans: Yes, we have clarified these notations. $D:\mathcal{L} \rightarrow \mathcal{M}$ and is defined on Page 3 in the revised version. \lambda is a positive integer and we used the top \lambda latent representations in geodesic interpolation. Both notations have been clarified in the main algorithm in the revised version.
>
> **11) p5, Section 3.2, it is not clear what s is and it has a different typesetting in the bullet points 1 and 2.**
>
> Ans: We have fixed this inconsistency in Section 3.
>
> **12) The transpose symbol can be confused with the iteration number in Figure 4.**
>
> Ans: We have now fixed this throughout the paper and used $\top$ to replace all transpose symbols.

---

> ### Author Response · Authors · 2020-11-19
> **Response to the reviewer (part 2/3)**
>
> **13) it is not clear where Proposition 1 is needed in practice.**
>
> Ans: Proposition 1 is not needed in setting up our method. However, it highlights a useful point that the linear operator we constructed on RKHS is equivalent to a highly non-linear operator in the input space. This can provide an intuition why our linear method yields competitive performance with other nonlinear models that also estimates the density on the latent space (e.g., 2S-VAE, Glow + AE)
>
> **Theoretical setting:**
> **1. Some concepts are not well defined. For example: The authors always talk about a well-behaved VAE. However, the authors do not define what they mean by well-behaved.**
>
> Ans: Yes, the use of well-behaved was colloquial which we have clarified in the text. Here, it simply means that the AE is one that produces a ‘smooth’ latent space while also minimizing the reconstruction error. Since this depends on the heterogeneity in the specific dataset, we do not characterize it mathematically but note that some recent results on regularized autoencoders study mechanisms via which a smooth latent space can be encouraged (see Ghosh et al., 2020). Some of these results led to our use of a stronger regularization on the autoencoders and empirically we find that it consistently improves the generation quality by an observable margin (shown in Appendix 1).
>
> **2. In addition, looking at Table 1, it seems that the proposed model is not systematically the best performing when the VAE is well behaved (the FID for vanilla VAEs are often better than the proposed model).**
>
> Ans: This is an important point. We acknowledge that our method is not essentially the best performing in some cases, although in general achieves comparable performance to other methods. However, we should also notice that on data with inherently high pixel-wise variance (e.g. neuroimaging data (shown in Figure 8) due the high dimensionality), methods such as VAE fail yet our method is still able to generate sensible images. Further, compared to other methods that learn an additional density estimator on top of trained autoencoders (2S-VAE, SRAE_{Glow}), our method has very little computational overhead in terms of training (Figure 9) while still has a comparable performance. This is the main message we wanted to convey. In addition, as requested, we have updated re-evaluated FIDs as well as collages of randomly generated samples for our method and the baseline models using the architecture developed by Ghosh et al. (2020) in Appendix 4 and 5. The results show that our method is quite competitive in terms of overall image quality and FID scores.
>
> **3. I am not sure what “true solution” means p6, end of Properties. It would be useful if the authors could give the intuition to make the reader understand what the true solution means and the need of the geodesic interpolation.**
>
> Ans: In the literature Honiene et al. (2011), at least a few proposals have described using a weighted linear interpolation to solve the pre-image problem. This can be viewed as a simple heuristic for the preimage problem. But notice that in our setup, the latent space is spherical, hence a simple linear interpolation will not result in a point on the hypersphere. To avoid this problem, we utilize geodesic interpolation. Unfortunately, geodesic interpolation does not have a closed form solution on the hypersphere unlike Euclidean spaces. Consequently, we use the recursive form proposed in Salehian et al. directly although other strategies can easily be plugged in. The convergence analysis in Salehian et al. showed that the recursive estimator to compute weighted mean converges to the ``true solution’’, which is the Frechet mean, and this is meaningful in our setting.
>
> **4. Some hypotheses are not verified, for example, a) is K_z invertible in practice (see Properties p6), b) is the experimental covariance is a good approximation for the true covariance.**
>
> Ans: Positive-definiteness of the limiting NTK for sphere-supported data is proved in (Jacot, 2020) in Appendix A.4, and due to our design choice, the sampling distribution of the proposed module is restricted to a hypersphere, by design. Therefore K_z is asymptotically invertible in terms of the number of units(n) used to estimate NTK. However, in practice K_z can be non-invertible due to finite samples (n) and potential numerical error, especially when the dimension of the latent space is small. A weak regularization coefficient is necessary in those cases (e.g. lambda = 1e-7). Sample covariance is a biased but consistent estimator of true covariance. And in practice, as we do not have access to the true covariance, it can only be approximated by sample covariance.

---

> ### Author Response · Authors · 2020-11-19
> **Response to the reviewer (part 3/3)**
>
> **5. In general, the algorithm seems to be explained twice with different degrees of precision at the bottom of p5 and in Figure 3 (which should be referred to in the text). The paper would benefit from merging both explanations.**
>
> Ans:  In the revised version, we have removed the redundant algorithm and instead gives a high level interpretation of the procedure in section 3.2.
>
> **8. It is not clear to me why the new sampling strategy would follow better the distribution of real samples. I think this should be better explained.**
>
> Ans: NTK theoretically corresponds to a trained infinitely-wide neural network, and therefore it spans a rich family of functions that allows the posterior to better fit the empirical distribution on latent space. Other methods such as fitting Gaussian mixtures in (Ghosh 2020) could work but may face difficulties if the set of Gaussian mixtures does not cover the distribution on the latent space.
>
> **9. Experimental setting: The experiments are described in a way that is not reproducible.**
>
> Ans: We have now provided a Jupyter notebook example to test on MNIST examples, as well as updated a detailed description of the architecture and hyperparameters we used in Table 1 in Appendix 7. We welcome suggestions or feedback. We will also share the codebase after anonymization if time permits.
>
> **1) We do not know which hyperparameters are chosen (for the AE and for the new development) and how they were chosen.**
>
> Ans: We have provided a detailed description of our training procedure and hyperparameter choices in Appendix 7. We used the same architecture for all reported scores in Table 1 (except for the GANs, which is not our development and have been optimized in terms of FID using neural architecture search.)
>
> **2) We also do not understand how the 4000 landmark points are selected.**
>
> Ans: The landmark points are chosen based on their ridge leverage scores according to the algorithm described in (Musco & Musco, 2017), which shows a consistent improvement over uniform sampling. It is discussed in the Nyström’s approximation part in Section 3.2. Since this is not the central argument in our work, we decided not to elaborate the algorithm in the main text to avoid confusion and instead requested the reader to refer to the original work. If suggested, we are happy to include more details here.
>
> **3) It is not clear what SRAE_rand_interp is.**
>
> Ans: We have included the definition in the table description, in this revision.
>
> **4) I do not understand how Figure 5 is done, is it 2D projection of the generated samples? If yes, with which dimension reduction method? (“wrapped on S^2” a bit above the figure is not clear).**
>
> Ans: We have now provided code/jupyter notebook example for MNIST. The landmark points are chosen according to the algorithm described in https://arxiv.org/abs/1605.07583. Figure 5 was plotted by using an autoencoder to encode MNIST digits to S^2. We have clarified this in the revision.
>
> **A better ablation study could be done:
> In the Supplementary, the authors say that they use a regularised AE (with spectral normalization) for the SRAE. However, the results of a regularised AE without the development of the authors are not shown.**
>
> Ans: We have provided quantitative and qualitative results for regularized/non-regularized AE based on the architecture used in (Ghosh, 2020) in this revision. Please refer to A.4 and A.5 for updated results.
>
> **The authors do not justify theoretically the need for selecting the top $\gamma$ samples. Therefore, it would be interesting to vary $\gamma$ and observe differences in the results.**
>
> Ans: We have now provided an ablation study on $\gamma$, and the results indicate that a larger gamma in general leads to higher FID (worse quality of generation). However, choosing a very small \gamma increases the risk of generating samples that too closely resemble the training samples. We choose $\gamma = 10$ for the experiments to balance two sides of the effect, yet in practice a better $\gamma$ can be chosen depending on the application.
>
> **In general, the results seem rather poor. The new method outperforms vanilla VAE or basic SRAE only in one of the three datasets for which we have quantitative results. The generative adversarial models outperform systematically the proposed model.**
>
> Ans: We have updated qualitative results on all datasets and recalculated the FID on CelebA for our model as well as all baseline models we compare to, based on the reviewer suggestions. The results show that our method is competitive with other generative methods both qualitatively and quantitatively. Also, on high-dimensional data, our method outperforms vanilla VAE significantly. However, the reviewer will see that even in the cases where our method does not outperform other methods, we still enjoy a very significant computational advantage (which is the main message here), as it only adds a small computational overhead to training a normal autoencoder.

---

### Author Response · Authors · 2020-11-14
**Response to all the reviewers**

We sincerely thank all the reviewers for their thorough and valuable feedback and the appreciation of the novelty in our work. We also appreciate the criticism on the clarity and reproducibility of our work. We have uploaded two jupyter notebooks accompanied with part of the codebase in the supplement. We will give individual responses to each reviewer, edit the paper based on the reviewers’ concerns, and upload the entire codebase shortly.

---

### Decision · Program_Chairs · 2021-01-07
**Final Decision**

**Decision:**

Reject

**Comment:**

All four reviewers were against accepting the paper. A major point shared by everyone was lack of clarity: this included its overall writing, its discussion toward prior work, and imprecise math to explain the ideas. The paper did improve quite a bit over its revisions. Whether this clarified all of the reviewers' understanding of the paper remains unclear. The work may ultimately need another cycle of reviews to assess its quality.

Another shared point are a number of recommended ablations in the experiments, as well as going through more comprehensively in the set of studied datasets (R3), effect of AE choices (R2), and alternatives to the geodesic (R1, R2).